# Regulation of Tissue Inflammation by 12-Lipoxygenases

**DOI:** 10.3390/biom11050717

**Published:** 2021-05-11

**Authors:** Abhishek Kulkarni, Jerry L. Nadler, Raghavendra G. Mirmira, Isabel Casimiro

**Affiliations:** 1Department of Medicine, The University of Chicago, Chicago, IL 60637, USA; abhikulkarni@medicine.bsd.uchicago.edu; 2Department of Medicine and Pharmacology, New York Medical College, Valhalla, NY 10595, USA; jnadler@nymc.edu

**Keywords:** 12-lipoxygenases, 12-LOXs, 12/15-lipoxygenase, 12/15-LOX, lipoxygenases, eicosanoids, inflammation

## Abstract

Lipoxygenases (LOXs) are lipid metabolizing enzymes that catalyze the di-oxygenation of polyunsaturated fatty acids to generate active eicosanoid products. 12-lipoxygenases (12-LOXs) primarily oxygenate the 12th carbon of its substrates. Many studies have demonstrated that 12-LOXs and their eicosanoid metabolite 12-hydroxyeicosatetraenoate (12-HETE), have significant pathological implications in inflammatory diseases. Increased level of 12-LOX activity promotes stress (both oxidative and endoplasmic reticulum)-mediated inflammation, leading to damage in these tissues. 12-LOXs are also associated with enhanced cellular migration of immune cells—a characteristic of several metabolic and autoimmune disorders. Genetic depletion or pharmacological inhibition of the enzyme in animal models of various diseases has shown to be protective against disease development and/or progression in animal models in the setting of diabetes, pulmonary, cardiovascular, and metabolic disease, suggesting a translational potential of targeting the enzyme for the treatment of several disorders. In this article, we review the role of 12-LOXs in the pathogenesis of several diseases in which chronic inflammation plays an underlying role.

## 1. Introduction

Inflammation is a conserved mechanism that serves as a defense against injurious stimuli, including invasion of pathogens, tissue injury, and intracellular damage signals [1]. Cells release a variety of factors, including histamines, prostaglandins, and bradykinin. These signals promote an acute inflammatory response, including changes to vascular permeability, with localized infiltration and accumulation of immune cells from the circulation, accompanied by the release of inflammatory mediators such as cytokines, chemokines, and eicosanoids. This cascade of inflammation is followed by tissue remodeling and a repair process to restore tissue health. Usually, this inflammatory response is completed upon the eradication of the pathogens or after tissue repair. However, if the process is not appropriately terminated or becomes uncontrolled, it can lead to a maladaptive chronic inflammatory state that contributes to irreversible tissue damage resulting in disease pathology. Chronic inflammation is known to be an underlying cause of several diseases, such as metabolic syndrome, type 1 and type 2 diabetes (T1D and T2D), non-alcoholic fatty liver disease (NAFLD), hypertension, cardiovascular disease (CVD), chronic kidney disease, neurodegenerative, and autoimmune diseases [2].

## 2. The Mammalian Lipoxygenases

Lipoxygenases (LOXs) collectively represent a family of enzymes that catalyze the oxygenation of cellular polyunsaturated fatty acids (PUFAs) to form eicosanoid metabolites that function in inflammatory pathways in an autocrine, paracrine, or endocrine fashion [3,4]. The LOX enzymes catalyze the oxidative metabolism of multiple PUFA substrates, including arachidonic acid (AA), dihomo-γ-linoleic acid (DLA), linolenic acid (LA), docosahexaenoic acid (DHA), and eicosapentaenoic acid (EPA) to generate a multitude of bioactive products involved in pro- and anti-inflammatory activities. The cellular events that occur during resolution of inflammation employ a conversion from pro-inflammatory metabolites to pro-resolving mediators, such as lipoxins and resolvins, to dampen the immune response (Figure 1) [5]. Arachidonic acid serves as a major precursor for a variety of eicosanoids with inflammatory properties including the product 12-HETE, which will be the focus of this review.

LOXs are expressed in all cell types of hematopoietic origin, in particular, platelets and leukocytes. Their nomenclature is based on the location at which they insert the oxygen on their fatty acid substrate [6,7]. 12-LOX catalyzes the oxygenation of the 12th carbon of arachidonic acid to 12-hydroperoxyeicosatetraenoate (12(S)-HPETE), which subsequently is reduced by glutathione peroxidase to a more stable analogous hydroxy compound 12-hydroxyeicosatetraenoate 12(S)-HETE, or simply 12-HETE [8,9,10]. In mice, there are seven functional LOX genes (*Alox5, Alox12, Alox12b, Alox15, Alox8, Aloxe3,* and *Aloxe12*), three or four functional genes in zebrafish (*alox5a, alox5b, alox12*, and a possible *alox15* orthologue) and six functional genes in humans (*ALOX5, ALOX12, ALOX12B, ALOX15, ALOX15B,* and *ALOXE3*). Their expression patterns vary by tissue distribution and cell type. 

The genomic distribution of the LOX genes shows conservation across higher species. All *Alox* genes in mice, except for *Alox5*, are located in the LOX cluster 1 and 2 of chromosome 11 [6]. Notably, these lipoxygenase clusters reside near other chemokine clusters, and the genetic proximity is conserved in humans. Whereas *ALOX5* is located on chromosome 10 in humans, the rest of the LOX genes are clustered in chromosome 17p13.1 near other genes involved in inflammation [11,12]. 

## 3. The 12/15-Lipoxygenase Pathway

### 3.1. Functional Orthologs of 12-LOX

Human LOX enzymes include 5-lipoxygenase (5-LOX), which result in the production of leukotrienes, as well as the 12-lipoxygenases and the 15-lipoxygenases. 12-lipoxygenase was the first identified mammalian lipoxygenase. It was discovered in human and bovine platelets in the mid-1970s [13,14,15]. Different isoforms of 12-lipoxygenase have been identified and named after the cells in which they were found: platelet, leukocyte, and epidermal type. However, this convention can be problematic as it is not always precise. For instance, the “platelet type 12-LOX” can be found in both platelets and skin of humans and mice, but is not expressed in porcine platelets [15,16]. To add more complexity to the nomenclature, there is significant species-specific variation in terms of the products formed by 12-LOX and 15-LOX. In this regard, the use of the appropriate nomenclature is imperative, as orthologs of the same gene may have different stereo-specificities in different species, leading to variable ratios of arachidonic acid oxygenation products. For example, the mouse 12-LOX enzyme encoded by the murine gene *Alox12* produces mostly exclusively 12-HETE and is referred to as “12-LOX.” It is also known as platelet-type 12-LOX in mice and humans, and its human ortholog is encoded by *ALOX12*. Both human and murine 12-LOX lead to the production of 12-HETE [17]. However, the LOX enzyme encoded by the murine gene *Alox15* results in the production of both 12-HETE and 15-HETE at a ratio of 6:1 from arachidonic acid, and has been referred to as leukocyte-type 12-LOX or simply “12/15-LOX” [18,19]. In humans, its ortholog is encoded by *ALOX15* and has been referred to as “15-LOX-1” or human reticulocyte type 12/15-LOX. Its activity favors the production of 15-HETE and leads to small amounts of 12-HETE in a ratio of 9:1 from arachidonic acid [20,21]. Other LOX enzymes include the human 15-LOX-2, which is an epidermis type of LOX that is phylogenetically related to murine epidermis type 8-LOX, while the R stereoisomers of 12-LOX (12R-LOX) are expressed by the epidermis in both human and mice and are encoded by *ALOX12B* and *Alox12b*, respectively (Table 1). Notably, the orthologs of different LOXs exhibit distinct clustering in humans, mice, and zebrafish, demonstrating conserved sequence similarity in these species (Figure 2). This review will focus on the 12-HETE producing 12-LOX and 12/15-LOX in mice, and 12-LOX and 15-LOX-1 in humans. Henceforth, we will be referring to all these enzyme homologs as 12-lipoxygenases or 12-LOXs [22,23].

### 3.2. The 12/15-LOX Signaling Pathway

12-LOXs exert many of its known effects via arachidonic acid metabolism. 12-HETE is a lipid molecule that can easily transit through cell membranes and induce its effects. Intracellularly, 12-HETE generation promotes oxidative stress, while extracellularly 12-HETE impacts a variety of signaling pathways to modulate inflammatory activity, possibly via interaction with G protein-coupled receptor 31 (GPR31) and its low affinity receptor BLT2 (Figure 3) [24,25,26].

### 3.3. 12-LOXs in Pancreatic Islet Inflammation and Autoimmune Diabetes

12-LOXs activity and 12-HETE levels have been linked to the pathogenesis of Type 1 diabetes (T1D). In T1D, autoimmune islet β-cell destruction leads to insulin deficiency. However, β-cell dysfunction prior to autoimmune destruction is a key early feature of the disease [27]. The non-obese diabetic (NOD) mice are used to study type 1 diabetes (T1D) owing to their spontaneous development of autoimmune diabetes. In this mouse model, impaired glucose-stimulated insulin secretion precedes the loss of β-cell mass by several weeks [28,29]. Strikingly, these mice are protected from the development of diabetes when *Alox15* is genetically deleted [11]. NOD mice are characterized by the presence of inflammatory cells within and around the pancreatic islets; a phenomenon referred to as insulitis [30]. In the absence of *Alox15*, NOD mice show reduced insulitis and maintenance of β-cell mass [31]. These findings were recapitulated using a specific 12-LOX inhibitor, ML351 [32]. However, neither of these studies identifies the cellular source of 12-LOX, leading to increased islet inflammation. In this regard, knockout of *Alox15* in mice on the C57BL/6 background protected against hyperglycemia and β-cell loss in response to the β-cell toxin streptozotocin (STZ) [33], suggesting that 12-LOX in islet β cells themselves may give rise to islet dysfunction. In support of this contention, protection from diabetes following STZ delivery was also seen in mice with a pancreas-specific knockout of *Alox15* [34]. In humans, immunostaining studies support the expression of 12-LOX in islets of donors at risk for T1D and in donors with more advanced T1D [35]. Unlike in islets of NOD mice, however, 12-LOX colocalizes with some types of pancreatic polypeptide-producing cells rather than β cells in humans. These rather striking findings suggest the possibility that 12-LOX expression might herald a trans-differentiation/de-differentiation event in β cells during the progression of the disease.

Modeling of islet inflammation using incubations with pro-inflammatory cytokines (PIC) has been performed in vitro using isolated islets (human and rodent) as well as β-cell-derived cell lines [36,37,38,39,40,41]. This PIC-mediated inflammation has been shown to increase levels of 12-LOX and its major product, 12-HETE [42,43]. Islet dysfunction under these conditions is likely to be mediated by the actions of 12-HETE, as human islets incubated with 12-HETE show reduced glucose-stimulated insulin secretion at lower concentrations (1nM) and induction of cell death at higher concentrations (100 nM) [43]. More importantly, in the presence of 12-LOX inhibitor ML355, the human islets treated with PIC or islets from T2D donors exhibit improved insulin secretion and oxygen consumption rate [44].

Mechanistically, 12-HETE appears to exert its detrimental effects on the islet through at least two complementary mechanisms. In the first, 12-HETE leads to the activation of NADPH oxidase-1 (NOX-1), which promotes oxidative stress in studies of both mouse and human islets [45]. In the second, 12-HETE appears to suppress the nuclear translocation of nuclear factor erythroid 2–related factor 2 (NRF2), the result of which prevents the protective antioxidant response [34]. The role of 12-LOX and its metabolite 12-HETE in the generation and propagation of oxidative damage in the islet has also been confirmed in zebrafish [46], thereby emphasizing the evolutionarily conserved role of this pathway in islet inflammation and damage. Beyond a role for 12-HETE in promoting oxidative damage to β cells, several studies emphasize that oxidative stress and endoplasmic reticulum stress might drive the production of neoantigens by β cells, thereby serving as triggers for the activation of adaptive immunity [47,48].

The cellular mechanisms by which 12-LOXs promote β cell dysfunction and death in T1D likely also involve a role for macrophages. Macrophages are among the earliest invading cells in T1D [49], and 12-LOX production by these cells is thought to promote a pro-inflammatory phenotype [50]. In recent studies, morpholino-based depletion of *alox12* in zebrafish resulted in an impairment in macrophage migration resulting from reductions in chemokine CXCR3 production [51]. Myeloid-specific deletion of *Alox15* in NOD mice led to reduced macrophage infiltration into the islet, preserved β-cell mass, and protection from diabetes [51]. The reduction in infiltrating T and B cells in this model likely reflects the important role for macrophages as “bridges” that connect β cells to the adaptive immune system, rather than any inherent role for 12-LOXs in adaptive immune cells. The expression of 12-LOXs in macrophages might account for the observation that circulating levels of 12-HETE in humans with new-onset T1D are elevated compared to control individuals [52]. Collectively, data from islet and macrophage studies suggest that signaling by 12-LOXs serve as a common pathway in both cell types that enables autoimmunity in the setting of T1D.

## 4. 12-LOXs in Insulin Resistance and Obesity

In obesity, tissue insulin resistance creates demand upon the β cell to enhance insulin release. T2D occurs when the demand exceeds the capacity of the β cell to meet insulin requirements, reflecting an inherent dysfunction of β cells in some individuals [53]. As macrophages accumulate in tissues (e.g., adipocytes, and islets) during obesity, they produce pro-inflammatory cytokines (e.g., TNF-α, IL-1β, and IFN-γ) that cause biochemical and physiological effects locally within the tissues and systemically, leading to insulin resistance [54]. Macrophage 12-LOXs are upregulated under conditions such as hyperglycemia and systemic inflammatory responses to promote macrophage migration and a pro-inflammatory state through the production of pro-inflammatory cytokines [42,50,55,56,57]. 12/15-LOX isoforms (encoded by human *ALOX12* and *ALOX15*) were highly expressed in adipose tissues from patients with obesity, particularly in the stromal vascular fraction (SVF), which contains the majority of inflammatory cells such as macrophages [58]. Furthermore, the addition of 12-HETE and 12-HPETE to adipocytes in culture promotes the expression of pro-inflammatory cytokines and chemokines, such as TNF-α, MCP-1, and IL-6.

When *Alox15^−/−^* mice are fed a high-fat diet (either a Western diet composed of ~45% kcal from fat or a high-fat diet composed of 60% kcal from fat), they show protection from impaired glucose and insulin resistance compared to similarly-challenged control mice [59,60]. Consistent with the reduction in insulin resistance, islets from these mice do not exhibit the adaptive hyperplasia seen in controls [59,60]. Adipose depots from dietary challenged *Alox15^−/−^* mice contain significantly less macrophage infiltration, consistent with either a potential role for 12-LOXs in macrophage migration or in adipose tissue chemokine secretion [59,60]. With regard to the latter possibility, studies from adipose-specific *Alox15* knockout mice (using aP2-Cre-driven deletion) also revealed improved glucose metabolism and reduced macrophage infiltration on a high-fat diet [61]. However, whereas the aP2-Cre transgene has been widely used to achieve pan-adipose gene deletion, *aP2* is also expressed in various tissues, including macrophages [62], and is involved in myeloid lineage differentiation during development [63].

A third key tissue that is involved in T2D pathogenesis is the islet β cell, which is also a site of activity by 12-LOXs. Deletion of murine *Alox15* in the pancreas (using Pdx1-Cre-mediated deletion) results in the loss of 12-LOXs primarily in β cells, since *Alox15* does not show substantial expression in other pancreatic cell types [34]. In these studies, pancreas-specific *Alox15* knockout mice showed protection from high-fat diet-induced glucose intolerance, with augmented adaptive islet hyperplasia and increased insulin levels [34]. These findings emphasize the potentially suppressive role of 12-LOXs and its products on β-cell function in response to insulin resistance. Taken together, studies on the perturbation of 12-LOXs in adipose, macrophages, and pancreatic islets emphasize a central role for their activity in the pathogenesis of insulin resistance and T2D.

## 5. 12-LOXs in Hepatic Inflammation and Disease

12-LOXs have been implicated in hepatic inflammation [64]. Hepatic inflammation can lead to a range of pathologies from non-alcoholic fatty liver disease (NAFLD) and non-alcoholic steatohepatitis (NASH) to alcoholic liver disease and ischemia-reperfusion injury. In these studies, roles of 12-LOXs encoded by both mouse *Alox12* and *Alox15* appear to be relevant. In one study, whole-body *Alox15^−/−^* mice were protected from hepatic steatosis under conditions of high-fat diet feeding. The infiltration by lymphocytes was diminished, pro-inflammatory cytokine mRNA levels (TNF-α and IFN-γ) were reduced, and there was a general reduction of chemoattractant mRNA levels, including that of Cxcl1 and Cxcl2/3 [65]. A potential role for 12-LOXs in NASH has also been reported. In a murine model of NASH, which involves the feeding of methionine and choline-deficient diet (MCD) [66], one study observed increased transcription of *Alox12* in hepatocytes and upregulation of serum 12-HETE levels [67]. An upregulation in both the *Alox12* mRNA and protein levels of 12-LOX in the liver was recently observed in these mice, along with enhanced activity of 12-HETE and 15-HETE oxygenation products in the cytosol of the hepatocytes [68]. It remains to be seen, however, if deletion of *Alox12* in this model protects against the development of NASH.

Hepatic ischemia-reperfusion injury is a disease process involving ischemia-mediated hepatocellular damage that is paradoxically exacerbated by subsequent liver reperfusion [69]. Levels of 12-LOX (encoded by *Alox12*) were found to be significantly upregulated in hepatocytes during the ischemic phase, leading to the accumulation of 12-HETE and to the production of pro-inflammatory mediators TNF-α, IL-6, and Cxcl2. The production of these pro-inflammatory mediators was shown to be mediated by a signaling axis involving the 12-HETE receptor GPR31 [70]. In a separate study, it was shown that the 12-LOX–12-HETE axis was activated in liver ischemia-reperfusion injury, and its activation was further enhanced in fatty liver. In this study, inhibition of 12-LOXs by the selective small-molecule inhibitor ML355 [71] mitigated the liver damage, and studies in vitro showed that 12-HETE increased the expression of GPR31 and activated the downstream PI3K/Akt/NF-κB pathway [72].

Finally, the role of 12-LOXs have also been studied in alcohol-induced fatty liver. Chronic alcohol consumption leads to ER and oxidative stress as well as liver injury in the form of fatty liver, alcoholic hepatitis, and cirrhosis [73]. Whole-body *Alox15^−/−^* mice were protected from alcohol-induced liver disease progression via suppression of oxidative and ER stress [74]. Alcohol feeding in this study led to an increase in reactive oxygen species generation in the liver of control mice, but was significantly reduced in *Alox15^−/−^* mice. Similarly, there was a marked reduction in classic ER stress markers IRE1α, ATF4, XBP1, and CHOP in *Alox15^−/−^* mice [74]. Collectively, these studies demonstrate that the 12-HETE-GPR31 pathway is activated in the setting of inflammation from both liver ischemia-reperfusion injury and NASH, while in the setting of alcohol induced liver disease, 12-LOXs play a pathogenic role through oxidative and ER stress pathways.

## 6. 12-LOXs in Gastrointestinal Inflammation and Autoimmune Disease

Eicosanoids produced from the conversion of arachidonic acid at the cell membrane have been shown to participate in host defense in the gastrointestinal system [75]. While the role of prostaglandins has been extensively studied in the pathogenesis of inflammatory bowel disease (IBD ), recent work has demonstrated that lipoxygenase products might play a role in autoimmune-related gut inflammation [75,76].

*Alox15* and its encoded 12-LOX is expressed in the intestinal epithelium [77]. Indeed, high levels of 12-HETE have been found in colonic mucosal tissue from patients with inflammatory bowel disease by thin-layer chromatography and high-performance liquid chromatography [76,78]. The neutrophil chemoattractant hepoxilin A3 (HXA_3_) is a downstream metabolite of 12-HPETE, and HXA_3_ functions in polymorphonuclear leukocyte (PMN) recruitment to sites of mucosal inflammation. Using a model that has been described to study basolateral to apical PMN transepithelial migration [79], it has been demonstrated that HXA_3_ promotes the final step of PMN recruitment to sites of inflammation by establishing a gradient across the epithelial tight junction. Inhibition of 12-LOXs by treatment with the small molecule inhibitor baicalein lead to blockage of HXA_3_ generation and inhibition of PMN transmigration stimulated by *Salmonella typhimurium* infection [80]. Furthermore, 12-LOXs have been shown to play a pathophysiological role in an animal model of IBD. In a dextran sodium sulfate (DSS) induced colitis model that was restricted to female mice, *Alox15^−/−^* mice were robustly protected from colitis and weight loss by a mechanism involving sustained epithelial tight junction protein expression [81]. Interestingly, expression of *Alox15* was not seen in healthy mouse colon but was significantly upregulated in the inflamed colon after 8 days of DSS induced colitis. Expression was restricted to the stroma cells, which represent invading leukocytes. Inflammatory marker analysis revealed that *Alox15* deficient mice exhibited less colonic macrophage infiltration assessed by F4/80 staining and mRNA analysis of distal colon revealed increase iNOS and TNF-α expression. Colon permeability studies revealed that *Alox15* deficient mice had significantly reduced permeability and higher functional ZO-1 expression compared to the DSS-treated control mice.

Thus, products of 12-LOXs appear to play a dual role in gut inflammatory disease by affecting gut epithelial cell integrity and by promoting polymorphonuclear leukocyte migration.

## 7. 12-LOXs and Cardiovascular Disease

Vascular remodeling is an active process in which the vessel wall thickens owing to vascular smooth muscle cell (VSMC) migration, thrombosis, and proliferation, leading to the formation of a thickened neointima layer in response to elevated shear stress, pressure, or arterial injury [82]. 12-LOXs have been implicated in VSMC migration, proliferation and apoptosis [83]. The metabolites 15-HETE and 12-HETE have both been found to act as mitogens in a MAPK dependent pathway in vascular endothelial cells and VSMCs [83,84]. Both 12-HETE and 15-HETE have been shown to promote VSMC migration via cAMP-response element-binding protein (CREB) mediated IL-6 expression [85].

Atherosclerotic lesions develop at sites of endothelial dysfunction and represent a chronic inflammatory process characterized by macrophage chemotaxis, accumulation, and foam cell formation [86]. Because of its ability to oxidize biomembranes and lipoproteins, 12-LOXs have been studied in the setting of atherogenesis. Human atherosclerotic lesions contain *ALOX15* mRNA and 12-LOX products [87,88]. 12-HETE has been shown to directly induce monocyte binding to human aortic endothelial cells [55,89], promote endothelial wall disruption [90], and to directly oxidize LDL, which contributes to foam cell formation [91,92]. In this regard, studies in which mice have been crossed with athero-susceptible backgrounds have shown protection from the development of atherosclerosis [93,94,95].

The most widely used mouse models of atherosclerosis are the *ApoE^−/−^ and Ldlr^−/−^* mice, which both develop hypercholesterolemia [96]. It has been demonstrated that *Alox15^−/−^* deficient mice on the *ApoE* null background developed significantly reduced atherosclerotic lesions even at 1 year of age. This observation was likely due to a reduction in oxidized LDL given the reductions in plasma autoantibody titers to oxidized LDL epitopes in double knockout mice compared to controls [95]. In another study, deletion of *Alox15* in the *Ldlr^−/−^* background also lead to a significant reduction in plaque formation at 3, 9, 12, and 18 weeks of high-fat diet feeding, while the cellular content of macrophages and T cells within the plaques did not change [93]. Both studies showed that *Alox15* deficiency did not influence lipid profiles. However, a study in which *Alox15^−/−^;Ldlr^−/−^* mice were fed a polyunsaturated fatty acid-enriched diet (10% calories as safflower oil) in which 12-LOX products are enhanced, atherosclerosis was reduced, but levels of cholesterol and triglyceride also decreased with improvement in hepatic steatosis compared to controls [97]. Therefore, 12-LOXs are involved in several steps in the pathogenesis of atherosclerosis, specifically through the promotion of LDL oxidation and induction of a pro-inflammatory state which promotes macrophage metabolic activity. Whether 12-LOXs products affect lipid profiles relevant to human disease has not been demonstrated.

## 8. 12-LOXs in Neuroinflammation and Neurodegenerative Disease

Neuroinflammation is an underlying cause of neuronal damage and brain disorders, including Alzheimer’s disease (AD) and Parkinson’s disease (PD) [98,99,100]. One of the main features of AD is the presence of senile plaques containing beta-amyloid peptide. 12-LOXs and 12-HETE are highly expressed in neurons of human brains [101], and 12-LOXs has been implicated in promoting neuroinflammation in both humans and mice [79]. Different mechanisms have been proposed for 12-LOXs mediated neuroinflammation, including its role in increasing oxidative stress in the neurons [101,102]. The transcription factor c-Jun is associated with neuronal apoptosis and has been shown to be active following exposure of neurons to the beta-amyloid peptide found in AD [103]. However, inhibition of 12-LOXs using anti-sense oligonucleotide strategy leads to a disruption in c-Jun dependent beta-amyloid-induced apoptosis in cortical cells [104]. Post-mortem analyses of brain sections from patients with AD show increased levels of 12-LOXs in the temporal and frontal lobes, with increased levels of both 12-HETE and 15-HETE [101]. Another study reported increased levels of 12-HETE in the cerebrospinal fluid of AD patients [105]. These studies implicate a role for 12-LOXs in the formation of beta-amyloid deposits. In a transgenic mouse model of AD-like amyloidosis in which mice develop amyloid deposits and cognitive impairment, mice were found to have a significant reduction in beta-amyloid production and deposition, as well as improvement in memory in the absence of *Alox15* [106].

Neuroinflammation is also considered as an underlying cause of the pathogenesis of Parkinson’s disease [107]. Although the pathogenesis of PD has focused on the presence of Lewy bodies in dopaminergic neurons, more recent studies have implicated oxidative stress and altered protein metabolism in precipitating the risk for PD development [107,108]. The 12-LOX pathway has been implicated with the progression of Parkinson’s disease through its role in oxidative and ER stress. An early finding of PD is a reduction in the level of the anti-oxidant glutathione [109], and its reduced levels lead to increased levels of nitric oxide (NO) in the neurons, which is neurotoxic [110]. An in vitro study with rat midbrain cultures showed that the NO-mediated neurotoxicity is reduced with the inhibition of 12-LOX with baicalein [111]. Moreover, studies in vitro with murine neurons show that a reduction in glutathione levels is associated with the upregulation of 12-LOX protein levels as well as 12-HETE [112].

12-LOXs and their pro-inflammatory mediators have also been implicated in nerve cell death (94) and brain ischemia [113,114]. In a murine model of transient middle cerebral artery occlusion, levels of 12-LOX were increased in the peri-infarct region of the neurons, and expression levels of 12-HETE also increased after brain ischemia in gerbil forebrains [115]. In a mouse study, when 12-LOX was inhibited by baicalein or genetically inactivated (*Alox15^−/−^*), mice were protected against transient focal ischemia [113]. A recent study showed that subarachnoid hemorrhage led to increased 12-LOX protein levels in murine brain macrophages and promoted neuronal death. When 12-LOX was inhibited by ML351 or by genetic inactivation (*Alox15^−/−^*), neuronal death was reduced, resulting in protection from brain edema and improved behavioral outcomes [116]. Treatment of mice with a novel inhibitor of 12-LOX, LOXBlock-1 reduced infarct sizes at both 24 h and 14 days post-stroke, with improved behavioral parameters. LOXBlock-1 also reduced oxidative stress in the cultured murine neuronal HT22 cells. It was observed that 12-LOX co-localized with lipids MDA2. This co-localization was also detected in the brain of two human stroke patients [114]. Indeed, plasma levels of 12-HETE were found to be elevated up to 7 days after stroke in a biomarker study that involved over 60 stroke patients [117]. Together, these data point to a critical role of 12-LOX in oxidative stress-related glutathione depletion in neuronal cell death relevant to human disease.

## 9. 12-LOXs in Pulmonary Inflammation and Disease

Pulmonary inflammation can be caused by infectious or non-infectious agents. Acute pulmonary inflammation leads to immune cell infiltration, mucus production, vascular leak into the airways, and epithelial cell damage. Unregulated inflammation is an underlying cause of many chronic pulmonary diseases such as asthma, chronic obstructive pulmonary disease, and fibrosis [118].

Eicosanoid levels are known to increase in response to inflammatory stimuli in the lungs [119], and PMN infiltration into the pulmonary space is a hallmark feature of pneumococcal pneumonia. While PMN activity is imperative to innate immunity, uncontrolled inflammation can result in tissue destruction and lung disease. In a model of *Streptococcus pneumoniae* infection, 12-LOXs play a major role in PMN migration to the site of pneumococcal infection. Indeed, inhibition of 12-LOX with baicalein prevented the transepithelial infiltration of PMN cells and reduced pulmonary infiltration. Furthermore, depletion of 12-LOX activity in *Alox15^−/−^* mice led to reduced bacteremia and increased survival as compared to controls where the pulmonary challenge with *S. pneumoniae* was lethal [120]. In an acute lung injury (ALI) mouse model, mouse inhalation of LPS led to induction of inflammation, increased vascular permeability, and upregulation of 12-LOXs. This study also revealed that depletion of 12-LOX lead to significantly reduced vascular permeability upon LPS treatment along with improved gas exchange and increased survival compared to the control littermates [121]. The same group further demonstrated that inflammation in this ALI model is mediated in part via recruitment of neutrophils, as depletion of 12-LOX significantly reduced neutrophil infiltration and prevented edema formation [122]. Therefore, these data demonstrate that 12-LOXs inflammatory activity is crucial in pulmonary infection pathology.

Non-epithelial lung cells appear to exhibit 12-LOX activity in relation to allergic airway inflammation [123,124]. A study showed that intranasal administration of 12-LOX to healthy Balb/c mice leads to airway epithelial injury that promotes airway hyper-responsiveness as seen in asthma. Production of 12-LOX in alveolar macrophages and fibroblasts leads to bronchial epithelial injury via 12-HETE in an IL-13 dependent mechanism [125]. In an allergen-induced lung inflammation model, depletion of 12-LOX (*Alox15*) reduced airway inflammation as seen by reduced bronchoalveolar lavage fluid leukocytes (eosinophils, lymphocytes, and macrophages), decreased cytokines (IL-4, IL-5, and IL-13), and reduced luminal mucus secretions in *Alox15^−/−^* mice compared to wt controls [126]. Recently, it was also shown that deficiency of *Alox15* impairs the granulopoiesis of neutrophils and prevents inflammatory responses to fungal *Aspergillus fumigatus* infection in the lungs [127]. In humans, a hypomethylation of *ALOX12* is associated with asthma in children [128]. Therefore, 12-LOXs participate in lung disease first by affecting infectious inflammation through PMN recruitment and second through promoting increased leukocyte cytokine production in airway hyper-responsiveness.

With respect to human disease, 12-LOXs and 12-HETE have been implicated in inflammatory lung disorders. Lungs of patients exposed to sulfur mustard toxin showed obstructive and restrictive lung disease and an increase in 12-LOX expression compared to control patients [129]. Furthermore, in tuberculosis, the expression of *ALOX12* increases, and it is positively correlated with neutrophil count and bacterial load in the airway [130]. Together these results implicate the 12-LOX pathway in airway epithelial injury relevant to human disease.

## 10. Role of 12-LOX in COVID-19

Coronavirus disease 2019 (COVID-19) is caused by the viral pathogen severe acute respiratory syndrome coronavirus 2 (SARS-CoV2), and its primary site of infection is the lungs. Although many cases of infection are asymptomatic or mild, severe cases of COVID-19 are characterized by bilobar pneumonia leading to acute respiratory distress syndrome, septic shock, and multi-organ failure [131]. Studies have shown that cytokine storm from an inflammatory response is involved in severe COVID-19 cases [132,133,134]. Notably, lipid analysis of bronchoalveolar lavage fluids from COVID-19 patients revealed that levels of 12-HETE and 15-HETE were significantly elevated compared to healthy controls [135]. Another recent study in which lipid mediators were analyzed from sera obtained from SARS-CoV-2-infected patients and healthy controls found that increased PUFA containing lipids were increased in infected patients, and this PUFA pattern was exacerbated with increasing COVID-19 disease severity. Moderate and severe disease was characterized by higher levels of 5-HETE and 12-HETE, suggesting that PUFA metabolites and, in particular, the 12-LOXs may play a paramount role in COVID-19 infection [136]. These data suggest a clear role in the systemic lipid biomolecule network in the pathogenesis of COVID-19. Upregulation of 12-HETE suggests that the 12-LOX pathway might be upstream of the cytokine storm which precedes alveolar damage. However, it still remains unknown whether activation of 12-LOXs is necessary for viral replication. More studies with 12-LOX inhibitors and 12-LOX genetic knockouts are necessary to (a) determine the precise mechanisms and pathways that are involved in this disease onset and progression, and (b) identify novel targets to prevent the progression of COVID-19 infection and/or alleviation of the associated alveolar damage.

## 11. Conclusions

PUFA metabolites play essential roles in cellular homeostasis and have been a topic of interest for the past several years. The 12-LOXs are lipid metabolizing enzymes that are expressed in all metabolically active tissues and regulate different cellular processes via their eicosanoid products, particularly 12-HETE. In homeostatic conditions, these eicosanoids have multiple functions including regulation of inflammation, platelet aggregation, cellular migration and modulation of vascular permeability. 12-LOXs have the ability to oxygenate several PUFAs into downstream mediators with pro- and anti-inflammatory effects. Loss of function models in which *Alox12* or *Alox15* have been deleted or 12-LOXs activity blunted, have highlighted a major role in pro-inflammatory activity associated with chronic disease. Although, a majority of studies demonstrate the pathological pro-inflammatory effects of 12-LOXs, it is important to recognize the anti-inflammatory role of the enzyme activity. Other substrates of 12-LOXs include DHA, EPA and LA (Figure 1). The omega-3 PUFAs DHA and EPA are the precursors of anti-inflammatory molecules such as resolvins and protectins [137,138]. Linoleic acid is the major PUFA in the Western diet serving as a precursor to AA. While AA is present in higher amounts in the body than LA; EPA, and DHA are found in much lower amounts than both AA and LA [139]. Interestingly, the consumption of omega 3 fatty acids has decreased significantly over the last 200 years [140], and diet has been shown to play a major role in which products are made [141,142,143]. Hence, in chronic inflammatory diseases, there are increased levels of arachidonic acid as a substrate [142,144,145], which could be the reason for the subsequent increase in the levels of pro-inflammatory eicosanoids like 12-HETE compared to other metabolites. Direct evidence exists linking 12-LOXs to several diseases characterized by unregulated inflammation, including diabetes (Type 1 and Type 2), atherosclerosis, obesity, neurodegenerative disorders, gut autoimmune disease, and pulmonary inflammatory disease including the recently identified SARS-CoV2 (Figure 4).

A major question that remains unanswered is whether the upregulation of 12-LOXs is the cause or the consequence of the disease. In either case, the circulating levels of 12-LOX eicosanoid products could serve as potential biomarkers for early detection of disease. Additional studies will need to be conducted to determine if targeting the enzyme, the dietary fatty acid modification, and/or blockade of the 12-HETE receptor will provide the optimal therapeutic advances. With respect to therapeutic targeting, several inhibitors are at the developmental stages for specific targeting of 12-LOXs [146]. It is intriguing that in mouse models, depletion or inactivation of 12-LOX prevents major disease development and/or progression, though it will be important to determine if there are deleterious global effects of inactivation of the 12-LOXs, as the *Alox15* null mice show hematopoietic defects leading to a myeloproliferative disorder as well as defective erythropoiesis [147,148]. Therefore, it is critical to follow these animal models for extended periods to ensure that there are no long-term adverse effects of 12-LOX inhibition. It is noteworthy that most of these findings have been limited to murine models. Hence, for the translation of these studies to clinical trials, it is essential to investigate the implication of 12-LOXs inhibition in human diseases. For this, studies with human-derived samples, including cell lines, induced pluripotent stem cells, primary cells, and tissues are necessary to verify the effects of 12-LOX inhibition and their effect on both 12-HETE and 15-HETE activity. In conclusion, based on the experimental evidence from rodent models, targeting 12-LOXs presents an attractive and promising strategy for treating inflammation-associated pathologies.

## Figures and Tables

**Figure 1 biomolecules-11-00717-f001:**
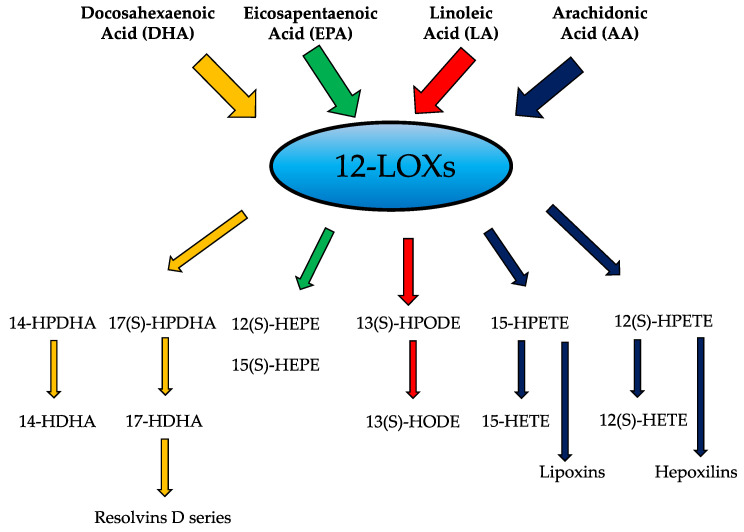
Substrates and products of the 12-lipoxygenases.

**Figure 2 biomolecules-11-00717-f002:**
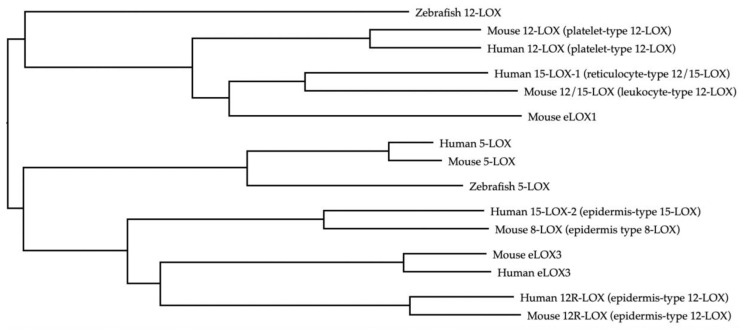
Lipoxygenase orthologs.

**Figure 3 biomolecules-11-00717-f003:**
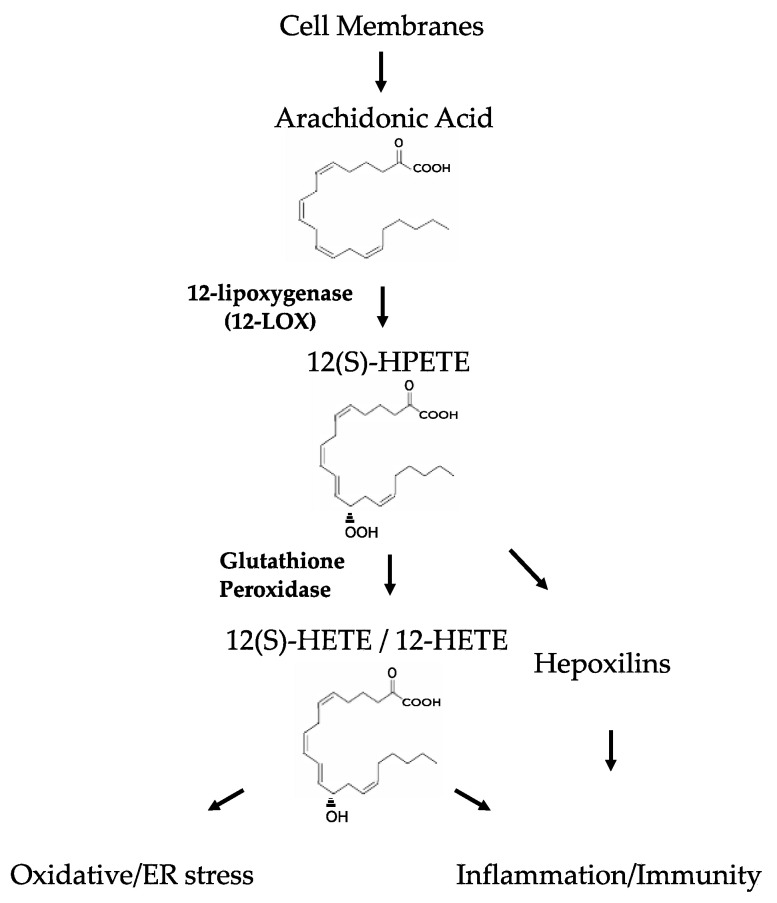
The 12-Lipoxygenase Pathway.

**Figure 4 biomolecules-11-00717-f004:**
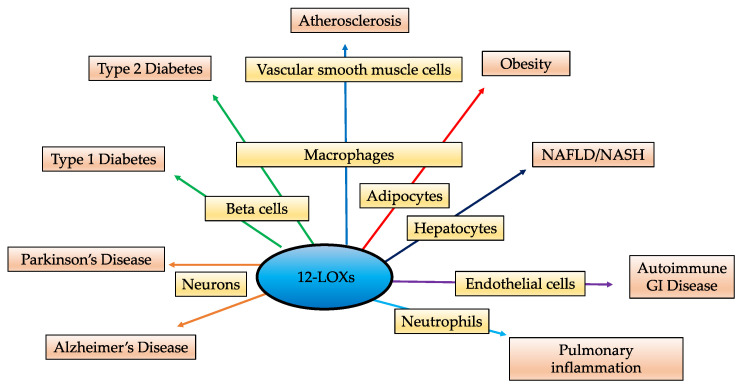
12-Lipoxygenases play a role in a variety of inflammatory pathways and present a potential target for several inflammatory conditions.

**Table 1 biomolecules-11-00717-t001:** LOX-encoding genes across species.

Human Gene Products	Mouse Gene Products	LOX Protein	Tissue Expression
*ALOX5* 5(S)-HPETE 5(S)-HETE	*Alox5* 5(S)-HPETE 5(S)-HETE	5-LOX	Leukocytes, macrophages, dendritic cells
*ALOX12* 12(S)-HPETE 12(S) HETE	*Alox12* 12(S)-HETE	Platelet type 12-LOX	Platelets, leukocytes, skin thrombocytes, β-cells (human)
*ALOX15* **15-HETE** 12(S) HETE	*Alox15* **12(S)-HETE**; 15(S)-HETE	Mice: 12-LO/Leukocyte-type 12-LOX/12/15-LOX Humans: 15-LO/15-LOX-1/reticulocyte type 12/15-LOX	β-cells (mouse), eosinophils, epithelial cells, macrophages (human, mouse), reticulocytes, smooth muscle, islets (mouse)
*ALOXE3* Hepoxilin	*Aloxe3* Hepoxilin	Epidermis-type LOX-3 (eLOX-3)	Skin
*ALOX12B* 12(R)-HPETE, 12(R)-HETE	*Alox12B* 12(R)-HPETE, 12(R)-HETE	12R-LOX	Epidermis
*ALOX15B* 15(S)-HPETE, 15(S)-HETE	*Alox8* 8(S)-HPETE 8(S)-HETE	Mice: 8-LOX Human: 15-LOX-2	Hair follicles, prostate, lung, cornea, macrophages (human)

Bold: main product produced.

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
