# Peer review of "Regulation of Tissue Inflammation by 12-Lipoxygenases"

_biomolecules, 2021, doi:10.3390/biom11050717_

Round 1

Reviewer 1 Report

This is a useful compendium and review of the potential involvement of 12-lipoxygenase(s) in various aspects of tissue inflammation.

My main point of critique is the befuddled distinction between the human 12-lipoxygenase (still usefully annotated across species as the platelet-type 12-LOX) and any other enzymes. As the authors are fully aware, humans only have one “12S-lipoxygenase” gene and expressed enzyme. So, what else is being considered here? If the animal “leukocyte-type” 12-LOX is to be included, then so should the human ALOX15, because that’s the equivalent human gene and enzyme. I was well into reading this review before becoming aware that a mixture of enzymes are included as “12-lipoxygenase”.

The authors really need to think about this because either the entire review has to be relabeled as “12-lipoxygenases” (plural), or stick to the human 12-LOX and its equivalent animal gene and gene product. If the leukocyte-type 12-LOX is to be included, then so should the human equivalent, ALOX15. On the other hand, if the review is centered on 12-HETE, then that’s another issue and the review should be retitled and make this the focus.

On reading the text again, everything states “12-lipoxygenase” – singular.

Minor points

Figure 1 – you mean EPA (not ETA) – same mistake on top line of page 2 – penta not tetra. The figure focusses on 12-LOX (singular), for which, according to Shozo Yamamoto (Takahashi Y) and others, LA is a very poor substrate, so including LA going to 13-HODE in the figure is misleading. (Or do you intend to included 12-lipoxygenases, plural, which is a completely different story?).

Figure 3 – this is strange. It is drawn as if a phylogenetic tree, but obviously all arms are equal so it is not a phylogenetic tree. Figure 3 is completely devoid of the comparative information that a phylogenetic tree displays and blurs the real relative relationships.

Page 3: you emphasize the mouse 12/15-LOX favors 12, and you really should note that the human 15-LOX is also a 12/15-LOX that makes about 10:1 in favor of 15 (this also applies to Table 1).

Page 3: Quote: “Thus, the mouse 12/15-lipoxygenase encoded by Alox15 may be more functionally related to the human platelet 12-LOX and the zebrafish 12-lipoxygenase enzyme encoded by alox12.”

The zebrafish issue, I have no knowledge of, but it is complete anathema to this reviewer to conclude that there is a functional equivalence of the leukocyte-type and platelet-type 12-LOX in human versus mouse. This does not stand up to the evidence or long-standing analyses of these enzymes. It would be reasonable to state that they both make 12-HETE, and that that could lead to common effects.

Table 1: 12R-HPETE is a substrate, not a product, of eLOX3.

There is considerable evidence that ALOX15B is expressed in monocytes.

Reviewer 2 Report

In their review the authors purport to discuss the current literature related to the biology of ALOX12 in the regulation of inflammation focusing primarily on the biology of one of its products 12-HETE. The major issue that I have with the review is that the primary evidence that is brought forward by the authors for their discussion relies on mouse data where the gene that is knocked out bares closer resemblance to the human ALOX15 than it does ALOX12. This aspect and the fact that this enzyme has a dual catalytic role in this reviewer’s opinion weakens the argument brought forward by the authors for a role of ALOX12 carrying a pathogenic role since the evidence being discussed does not used ALOX12 KO mice but rather ALOX15 KO mice. It would have potentially been better for the authors to focus on the product ie 12-HETE or HXA3, given that there is clear biology here and would overcome the issue of species differences in enzyme catalytic function.

Furthermore, while this reviewer appreciates that the interest of these authors is in the pathogenic biology associated with this enzyme (or rather one of its products 12-HETE), as this is a review it is important that it provides a balanced discussion of the biology of this protein. Therefore, the authors should also include the role of this enzyme in the production of pro-resolving mediators and the protective biology described for these molecules. This is particularly important because for every biological action that the authors review in their manuscript there is ample literature that ALOX12 and ALOX15 derived pro-resolving mediators exert protective actions on.

Another aspect that the authors appear to have overlooked which is important is the fact that ALOX15 deficient mice are reported to display a myeloproliferative phenotype PMID: 17043146 and defective erythropoiesis PMID: 32918502. Therefore the statement in the abstract : ‘Interestingly, apart from disease prevention, these animal models do not exhibit detrimental symptoms from depletion or inhibition of 12-LOX, suggesting a translational potential of targeting the enzyme for the treatment of several disorders’ is incorrect and needs to be revised.

The authors should also note that human LOX 12 produces negligible amounts of 17S-HDHA and therefore figure 1 is incorrect, and should be revised.

The statement that human LOX-15 produces only 15-HETE from AA is incorrect. Indeed LOX-15 also produces a small amount of 12-LOX and has recently been demonstrated to produce 14Hp-DHA (PMID: 32324389). The authors should revise their statement and also provide the appropriate references.

Round 2

Reviewer 1 Report

All original points well addressed.

Reviewer 2 Report

The authors have overall addressed the comments raised